

# Linking the distribution of microbial deposits from the Great Salt Lake (Utah, USA) to tectonic and climatic processes

Anthony Bouton[1], Emmanuelle Vennin[1], Julien Boulle[1], Aurélie Pace[2], Raphaël Bourillot[2], Christophe Thomazo[1], Arnaud Brayard[1], Guy Désaubliaux[3], Tomasz Goslar[4,5], Yusuke Yokoyama[6], Christophe Dupraz[7] and Pieter T. Visscher[8]

[1] Laboratoire Biogéosciences UMR 6282 UBFC/CNRS, Univ. Bourgogne Franche-Comté, 6 boulevard Gabriel, Dijon 21000, France
[2] Géoressources et Environnement, Ensegid, Institut Polytechnique de Bordeaux, EA 4592, Université de Bordeaux, 1 Allée Daguin, 33607 Pessac, France
[3] GDF Suez, Exploration Production International, 1 place Samuel de Champlain, Faubourg de l'Arche, 92930 Paris La Défense Cedex, France
[4] Adam Mickiewicz University, Faculty of Physics, Poznań, Poland
[5] Poznań Radiocarbon Laboratory, Foundation of the Adam Mickiewicz University, Poznań, Poland
[6] Atmosphere and Ocean Research Institute, Department of Earth and Planetary Sciences, University of Tokyo, 5-1-5 Kashiwanoha, Chiba 277-8564, Japan
[7] Department of Geological Sciences, Stockholm University, Svante Arrhenius väg 8, Stockholm, 06269, Sweden
[8] Department of Marine Sciences, University of Connecticut, 1080 Shennecossett Road, Groton, CT 06340, USA

*Correspondence to*: Anthony Bouton (anthony.bouton@u-bourgogne.fr)

**Abstract.** The Great Salt Lake is a modern hypersaline system in which an extended modern and ancient microbial sedimentary system has developed. Detailed mapping based on aerial images and field observations can be used to identify non-random distribution patterns of microbial deposits, such as paleoshorelines associated with extensive polygons or fault-parallel alignments. Although it has been inferred that climatic changes controlling the lake level fluctuations explain the distribution of paleoshorelines and polygons, straight microbial deposit alignments may underline a normal fault system parallel to the Wasatch Front. This study is based on observations over a dm to km spatial range, resulting in an integrated conceptual model for the controls on the distribution of the microbial deposits. The morphology, size and distribution of these deposits result mainly from environmental changes (i.e. seasonal to long-term water level fluctuations, particular geomorphological heritage, fault-induced processes, groundwater seepage) and have the potential to bring further insights into the reconstruction of paleoenvironments and paleoclimatic changes through time. New radiocarbon ages obtained on each microbial macrofabrics described in this study improve the chronological framework and question the lake level variations that are commonly assumed.



# 1 Introduction

Lacustrine microbialites have been linked to significant petroleum systems, especially prolific ones in the Lower Cretaceous "pre-salt" systems found offshore in Brazil and Angola (Davison, 2007; Mello et al., 2012). The discovery of extensive microbial deposits associated with petroleum reservoirs has increased the interest in modern and ancient microbialite analogs

(Chidsey et al., 2015). The Great Salt Lake (GSL) shares numerous comparable geodynamical characteristics with pre-salt deposits (e.g. normal fault system framework, volcanic activity; Liu, 2001) and additionally shows a high diversity and abundance of lacustrine microbialites (e.g. Eardley, 1938). The GSL thus represents an appropriate modern analog for Lower Cretaceous and other petroleum systems, as well as microbial dominated lakes in general.

The GSL represents the last 11.5 ka BP phase of a long spanning lake history initiated ca. 30 ka BP ago with the Lake

Bonneville (LB) phase (Oviatt et al., 1992; Godsey et al., 2005). The first investigations into the GSL sedimentary record were performed by Eardley (1938) who extensively described its sediment lithology and distribution, including that of microbial deposits, referred to as calcareous algal bioherms. Recent geophysical methods have expanded the knowledge on these microbial deposits and provided a preliminary understanding of the factors controlling their spatio-temporal distribution, especially the active tectonic and anthropogenic effects (Baskin et al., 2011, 2012, 2013). Other studies have

paved the way to describe the GSL microbial system in comprehensive and integrated models and focused on (i) how the growth pattern of the microbial deposits was influenced by the geometry of the underlying substrate (Carozzi, 1962), (ii) the calcium carbonate precipitation processes (Halley, 1977; Post, 1980; Pedone and Folk, 1996), (iii) the petrography of the microbial deposits (Chidsey et al., 2015), and (iv) the adaptation of microbial organisms to this unstable and harsh environment (Baxter et al., 2005). However, the combination of detailed descriptions of both microbial and sedimentary

structures and the evaluation of the factors controlling their distribution through time (e.g. water level fluctuations, tectonics) have been poorly investigated to date.

In the GSL environment, microbial deposits are composed of both microbial mats and microbialites. Microbial mats are organosedimentary structures, characterized by the presence of phototrophic bacteria and high microbial metabolic rates (Stal et al., 1985, Van Gemerden, 1993). They form sedimentary ecosystems that consist of a diverse microbial community

embedded in an organic biofilm matrix (Decho et al., 2000, Ley et al., 2006; Baumgartner et al., 2009). Microbial mats can mineralize either through trapping and binding detrital sediments, and/or through microbially-mediated precipitation (Burne and Moore, 1987; Dupraz et al., 2009). The resulting structures, called microbialites, can be preserved in the rock record (Dupraz et al., 2011).

Here, we focused on the western margin of Antelope Island, in the southeastern part of the GSL (Fig. 1). We mapped the

spatial-temporal distribution of the microbial and non-microbial deposits as well as the sedimentary structures. Even if other microbial deposits ("tufa") have already been described for the Pleistocene LB (Felton et al., 2006), we focused on the well-developed and extended structures outcropping close to the modern-day water level. We relate the observed variations to the potential environmental (external) controlling parameters (i.e. bathymetry, shoreline migration, desiccation, hydrogeology).



Finally, we discuss the influence of climate vs. tectonics on the formation, distribution and preservation of microbial deposits through time.

## 2 Settings

The GSL is an endorheic hypersaline lake located in northwestern Utah (USA; Fig. 1). It is part of the Basin and Range
Province, which formed by the gravitational collapse of the North American Cordillera (Liu, 2001). The GSL surface area averages 4480 km² (since at least 1876) with a maximal depth of 10 m (Baskin and Allen, 2005). The mean historical water level (MHWL) has been set at 1280 m asl (above sea level; from the GSL level record available from the U.S. Geological Survey), which is used as the reference value in this study. The average salinity in the south arm of the GSL where Antelope Island is located (Fig. 1) is approximately 120 g L$^{-1}$ (from 1966 onwards), but shows an important range of fluctuations
(from 50 g L$^{-1}$ to 285 g L$^{-1}$ since 1966) resulting from the variations in the water volume (Rupke and McDonald, 2012). GSL water inputs come from rivers (66%; Bear, Jordan and Weber/Ogden Rivers), rainfall (31%) and subsurface flow (3%; Gwynn, 1996). As a closed basin lake, the GSL water level reflects the balance between these inputs and net evaporation. At an annual scale, the seasonality of the inputs and evaporation induces an average of 0.3-0.6 m fluctuations in the lake level (Chidsey et al., 2015). At a longer timescale, the GSL water level is mainly under the control of pluri-metre scale climate-
induced variations (Lall and Mann, 1995) resulting from the interplay between the climatic Pacific Quasidecadal Oscillation and the Interdecadal Pacific Oscillation (Wang and Gillies, 2012).

Modern fluctuations in the GSL level are in the same range as those estimated for its entire history (e.g. Murchison, 1989). However, the GSL only dates back to the last 11.5 ka BP, and is the remnant of the LB phase (from ca. 30 to 11.5 ka BP; Oviatt et al., 1992; Godsey et al., 2005) during which the fluctuations in the lake level were significantly higher. LB reached
a maximal surface area of 52,000 km² with a corresponding maximal depth close to 372 m (Patrickson et al., 2010). During this period, the mass of water was considered as sufficient (ca. $1.10^{16}$ m$^3$) to isostatically weigh down the crust of the earth thereby inducing a depression (Oviatt, 2015). The different water level falls following the LB highstand episode induce the consequent removal of the water load and generate an isostatic rebound of the crust to its original position (Oviatt, 2015). Therefore, the isostatic rebound differently affects the four main terraces, which are easily recognizable in the current
landscape (Stansbury, Bonneville, Provo and Gilbert; Patrickson et al., 2010), and which reflect the main lake level variations. Oviatt (2015) considered that the entire water level elevation below an altitude of 1300 m asl (i.e. at or below the Gilbert shoreline) was not affected by the isostatic rebound and suggest that the post-Bonneville rebound ceased before the GSL phase ca. 11.5 ka BP (around 13 cal ka).



## 3 Antelope Island: shore to lake transect and dating

In this study, we mapped the distribution of ancient and recent microbial deposits, non-microbial deposits and sedimentary structures along the GSL shoreline on the western margin of Antelope Island (Fig. 2). The shoreline terminology is used to describe the boundary line between the lake water and the land. Therefore, it is an ephemeral feature that constantly shifts in

the vertical and horizontal positions following changes in weather, climate, tectonics, and geomorphic processes (Oviatt, 2014). The shore is considered as the land along the edge of the lake and thus corresponds to an area that is usually emersed. In order to estimate the duration of microbialite development and improve the chronology of the lake level variations, nine new radiocarbon ages of microbial structures and ooids are provided in this study. Ages were measured using AMS spectrometers from the Atmosphere and Ocean Research Institute at the University of Tokyo (Yamane et al., 2014; sample

preparation and analytical details are different depending on the sample size and are described in detail in Yokoyama et al., 2007, 2010) and the Adam Mickiewicz University in Poznań (Goslar et al., 2004; the samples were prepared in the Poznań Radiocarbon Laboratory before the analysis).

### 3.1 Microbial deposits

Microbial deposits extensively colonize the lake margin and show a heterogeneous spatial distribution and macrofabrics

(Fig. 2). Five macrofabrics can be distinguished along the shore-to-lake transect distributed both on emersed or recent submersed area. On the emersed shore, fossil microbialites (Fig. 3a, b), composed of carbonate minerals (aragonite and calcite, associated with dolomite in pore spaces), are common but there is a lack of living microbial mats. In the submersed area, microbially-mediated precipitation (Dupraz et al., 2009) yields abundant microbialites (Fig. 3c, d) composed of carbonate minerals (aragonite evolving to dolomite) and preserved Mg-clays through microbially-mediated and early

diagenetic processes (Pace et al., ongoing work). Trapping and binding are involved in the accretion of these microbialites, which comprise living microbial mats. The mats show a succession of thin orange and thick porous green clotted laminae.

The fossil microbialites correspond to laminated and cauliflower structures (Fig. 2), with early phases dated ca. 21.9 ka BP (Table 1). The laminated and cauliflower structures (15 and 30 cm high) are subdivided in four growing phases: (1) a first phase of coalescent knobs, 1 to 2 cm thick, showing successive clotted-thrombolitic and mm-laminated microstructures; (2) a

second, 4–5 cm thick thrombolitic layer composed of peloidal and clotted clusters; (3) a third, 1–2 cm thick homogeneous laminae organized in a dense set of mm-large micro-columns; and (4) a fourth, 5–15 cm thick bushy structures composed of highly recrystallized vertical loose-clotted columns. The last bushy structures fused building the cauliflower appearance (Fig. 3e). All the phases extend laterally in a covering drape of in place Cambrian quartzite and blocks of cemented microconglomerates (Fig. 3b).

In the temporarily exposed zone of the shore, cow-pie structures are prevailing as individual and coalescent structures (Fig. 2), with early phases dated ca. 10.6 ka BP and up to present for living ones (Table 1). They are composed of microbialites, made of alternating laminated and clotted mesofabrics, covered by an active surface microbial mat when located below or



just few above the water surface. Semi-lithified to non-lithified cow-pie structures form an extended flat of coalescing structures with a thickness comprised between 2 cm to 30 cm. Each structure is limited to few dm to m width while coalescent structures colonizing the flat reach tens or hundreds of metres; (Fig. 3a). When coalescent, the merging structures form a linear growing upward ridge. The central part of cow-pie can be frequently truncated in the exposed area with an

5 erosion surface locally covered by a new cow-pie growing phase. The active microbial mat is also observed laterally as a flat pustular crust covering ooids sand ridges and sands as well as micro-conglomerates (Fig. 2). The associated mineralized microbially-mediated carbonates are dated ca. 0.064 ka BP (Table 1). Oncoid sands, resulting from the snatching of microbial mats, are observed accumulated along the shoreline (Fig. 2) and are considered as recent sediment (ca. -0.011 ka BP; Table 1). They show a clotted mesofabric similar to the one observed in the living mats.

In deeper environment, domes and columns (Fig. 2) present circular cross-sections (0.2-1.2 m in diameter). The top of one of them, below the active microbial mat, has been dated ca. 2.7 ka BP (Table 1). Temporarily exposed structures show deposition relief of less than 15 cm, but reach several decimetres-scale relief lakeward in deeper environment (1–2 m). Their mesofabric is mainly clotted with rare alternating laminated layers.

### 3.2 Non-microbial deposits

The non-microbial deposits present along Antelope Island western margin are organized in a shore to lake transect. The shore flat deposits are dominated by sub-angular to rounded boulders composed of Cambrian quartzite in the NW part of the island. At the 1280 m asl shoreline, the pluri-decimetric to metric-sized rounded boulders are aligned in a conglomeratic belt (Fig. 3a). This shoreline marks the distal boundary of boulders dominated area. The shore domain also locally exhibits metric- to pluri-metric sized tilted blocks of microconglomerates (Fig. 3b). These deposits are composed of angular quartz

grains, forming a micro-breccia. The grain sizes range from < 1 mm to more than 2 cm and are organized in planar beds of fairly sorted grains. The cementation is ensured by a fibrous aragonitic cement. Both boulders and microconglomeratic blocks are encrusted by the fossil laminated and cauliflower structures. Below 1280 m asl, ooid sands are prevalent along Antelope Island western margin (Fig. 3d). These sands are mainly composed of individual ooids associated with fecal pellets of *Artemia* and clastic grains as secondary components. The ooids sands can be classified into medium sands since the

diameter of the ooids mostly range between 200 and 500 µm. The ooids show alternating radial and concentric layers of aragonite forming the cortices. A bulk radiocarbon dating on the cortex indicates an age of 3.3 ka BP (Table 1). The nuclei are mostly made of fecal pellets or quartz fragments, even though some ooids have grown over ancient ooids or microbialite fragments. These sands are locally cemented, forming grainstones. The thickness of ooid sands is highly variable. In the shore flat, ancient ooid sands are preserved as 50 cm thick and 2–4 m large ridges parallel to the shoreline. In the temporarily

exposed part of the shore, ooid sands thickness evolve from few cm near the 1280 m shoreline to more than 70 cm in the ooid sand embayment of White Rock Bay (Fig. 1b). Deepward, the ooid sands overlie a finely laminated green to grey clay sediment, mostly composed of quartz, smectite, illite, and ca. 30% of carbonates (dolomite and calcite). In the exposed embayments, these clays were also observed in surface as 10–30 cm wide strips showing a 5 cm relief above the flat. These



strips form a polygonal network extending over hundreds of metres, individual polygon ranging from 5 to 20 m in diameter (Fig. 4a-b). In section, clay sediments exhibit a wedge-shape morphology widening toward the surface associated with gypsum crystals (Fig. 4c). Thin crusts of microbial mats can also be observed on surface of the clay strips (Fig. 4d). The microbial mats show green or red pigments (Fig. 4e-f), and evolve from a planar to hemispheroid uncemented fabric. Other

evaporites were also observed such as <1mm crust of halite at the surface of the exposed ooids sands and ephemeral of mirabilite ($Na_2SO_4 \cdot 10(H_2O)$) crystals in small depressions of the flat, frequently adjacent to the polygons (Fig. 4g-h). The sediments in the deepest parts of the lake consist of clays with a high amount of carbonates, especially near Antelope Island where Eardley (1938) reported a ca. 70% carbonate content (aragonite and dolomite).

## 4 Mapping of microbial and non-microbial deposits and sedimentary structures

**4.1 Method**

Geological field mapping (GPS, facies, sampling etc.) was performed during three field campaigns in June 2013, September 2013 and November 2015 on the northwestern shore of Antelope Island, extending from White Rock Bay up to Bridger Bay (Fig. 1). Field observations were combined with aerial and satellite images. Aerial images were obtained from Google Earth Pro 7.1.2 (from State of Utah, USDA Farm Service Agency and NASA; unknown remote-sensors), the Utah Automated

Geographic Reference Center (2014 NAIP 1 Meter 4-bands (RGB and infrared) Orthophotography, used in 3-bands RGB natural colour) and the USGS EarthExplorer (aerial photo single frames, NAPP, NAHP; mixing black and white, RGB and infrared bands). Satellite images correspond to Digitalglobe® images (WV02 spacecraft, Imaging bands Pan-MS1-MS2; obtained from Garmin Birdseye) and Landsat images (NASA Landsat Program, 1972 to 2015, L1-5 MSS/L4-5 TM/L7 ETM+ SLC-On/L7 ETM+ SLCOff/L8 OLI/TIRS, Sioux Falls, USGS, 08/07/72-10/13/2015). The dark orange to

dark green pigments of the modern living mats in the aerial and satellite images allow us to distinguish between submersed living microbial deposits and light coloured uncolonized ooid sands or clays. This mapping approach required suitable visibility through the atmosphere and water column, and images showing water turbidity, cloud cover or waves were discarded. Aerial images have been imported in ArcGIS in order to perform the mapping. The first approach consists in converting the pixel showing microbial deposits into black pixel and integrated them into a shapefile layer. The percentage

of points (i.e. microbial deposits) per surface unit was used in order to calculate the density of the microbial structures (in ArcGIS; Point density tool, 15 m output cells and 50 m circular neighbourhood). Calculations were difficult below a pixel resolution of 15 m due to the high amount of point; thereby it was completed with a visual approach improving the mapping resolution. For this we defined high and low densities areas of microbial deposits in aerial images which match with the perception in the field. These results have been emplaced in a metric topographic map for the GSL floor. This map was

constructed by extrapolating the feet-unit map provided by Baskin and Allen (2005).



## 4.2 Results

The distribution of the microbial, non-microbial deposits and relevant sedimentary structures along Antelope Island is mapped in Figs. 5 (standard resolution maps and aerial images) and S1 (high-resolution distribution map). It complements the maps published by Eardley (1938) and Baskin et al. (2012; based on geophysical data). Using recent aerial images of the Antelope Island area, we estimate that the coverage of the microbial deposits may be at least 90 km² wide with a patchy distribution and a density varying from isolated to a local full coverage of the lake bottom (Fig. 5a-b). A comparison of the temporal series of aerial images (since 1950) provides insights on the migration and burying dynamics of the microbial deposits (Fig. 6a-b). For instance, on the western Antelope Island margin, microbial deposits appeared to be intermittently buried by loose sediments (Fig. 6a-b). This process likely results in a ca. 10% underestimation of the total surface of the western Antelope Island margin covered by microbial deposits. An initial approximation of 260 km² has been proposed by Eardley (1938) for the extent of the microbial structures in the whole lake. Only considering the Antelope Island area, our mapping results reach a third of the total extent proposed by Eardley (1938). Therefore, its initial estimation is likely an underestimation as (i) a significant part of these structures can be buried under the ooid sand or clay sediments, and (ii) they are partially eroded around the shoreline (Fig. 6).

Most microbial deposits around the margin of Antelope Island were found between 7.5 m below (1272.5 m asl) and 3 m above (1283 m asl; Fig. 5c) the MHWL. Above 1280 m asl, microbial deposits are scattered and mainly correspond to fossil laminated and cauliflower structures. In the temporarily exposed zone of the shore facing steep topographic area, cow-pie structures are preserved along the flat. They are absent in the embayment areas (like White Rock and Bridger bays; Fig. 1) emplaced in front of low topographic area. They form a dense network just below 1280 m asl showing a scattered to a full coverage distribution. In addition, the cow-pie macrofabrics are located along linear belts following isobaths and are locally observed in a relative paleohigh induced by the presence of unusual conglomeratic deposits (Fig. 5h). Lakeward, cow-pie structures pass to domes and columns, which are organized as isolated circular structures or coalescent merging into clusters. Between 1276 m and 1278.5 m asl (4 m and 1.5 m below the MHWL), microbial structures are frequently arranged in m- to km-long linear belts following the isobaths and parallel to the shoreline (Fig. 5a, d). They are also organized following the edge of 20–80 m wide submersed polygons (Fig. 5a, e, h). Locally, the centre of the polygons can be colonized as well. The polygons are interconnected in a network covering a surface of tens of km² down to 1274 m asl. Deeper occurrences of microbialites were reported (e.g. Colman et al., 2002); they are visible on aerial/satellite images when the conditions (e.g. turbidity) are optimal (Fig. 5a, dashed yellow polygons; ca. 9 m below the MHWL, 1271 m asl). The depth limit of the microbial deposits along the western margin of Antelope Island (approximately 5 m) coincided with a relative sharp topographic drop off (Fig. 5a, c, f). This abrupt slope break marks the transition from a shallow platform shoreward with abundant microbial deposits, to a deeper area with seldom occurrences of microbialite (Fig. 5a, c). Along the slope break, scattered lens-shaped detachments of microbial structures have been observed (Fig. 5g).



## 5 Influence of water level fluctuations

### 5.1 Markers of the present-day water level

Living microbial mats develop on top of microbialites and cemented ooid sands documented in the submersed area. They can also persist on the shore where emersion is limited to a short time, e.g. corresponding to annual water level fluctuations.
Therefore, the upper occurrence of the living mat, corresponding to the boundary between living and ancient structures, is a good indicator of the annual mean elevation of the shoreline.

Onshore, the polygonal networks are associated with green clays and prismatic gypsum crystals, thin crusts of microbial mats resulting from blistering and mirabilite mineral reflecting alternating wet/dry conditions. Such conditions are suitable for the formation of desiccation polygons (Warren, 2006). The extended polygonal networks observed in the embayments area are indeed related to desiccation processes. The polygons are locally encrusted by microbial mats probably associated with fluid circulations in the underlying crack system (Bouton et al., in press). Both the sedimentary structures and the evaporite can be used to define the emersion conditions and to estimate the shoreline position. Based on a uniformitarianism approach, we therefore consider that similar structures with the same underlying processes are preserved in the fossil record.

### 5.2 Markers of past lake levels

The occurrence of ca. 21.6 ka BP laminated and cauliflower microbialites, up to 3 m above the MHWL, indicates that the lake level was (i) much higher when these structures developed, and (ii) lasted long enough to support mineral precipitation and growth. The growth rates reported in the literature are generally below 1 mm yr$^{-1}$ (e.g. between 0.1 and 0.54 mm yr$^{-1}$ for the Shark Bay microbialites, and potentially up to 0.75 mm yr$^{-1}$, Jahnert and Collins, 2012; ca. 0.16 mm yr$^{-1}$ in a Bahamian lagoon, with a maximal growth rate estimated at 0.88 mm yr$^{-1}$, Paull et al., 1992; and ca. 0.1 mm yr$^{-1}$ for the hypersaline dolomite stromatolites of Lagoa Salgada, Brazil, Bahniuk Rumbelsperger, 2013). Some of the laminated and cauliflower structures on the shore of the GSL are > 30 cm high. Continuous intervals of up to 15 cm and the absence of major discontinuities suggest no long-term exposures during microbial deposit growth. Considering a growth rate of 0.88 mm yr$^{-1}$ and 15 cm of continuous growth, the formation of these structures would have required at least 170 years.

The fig. 5 shows the distribution of the microbial deposits and associated features (faults, dominant sediment, GSL bottom topography etc.) along the western Antelope Island shoreline. Along a shore to lake transect, microbialites are frequently observed aligned and following polygons. Most of the cow-pie structures are aligned along the major 1280 m shoreline. They are observed in surface of proximal domes and columns alignments, alignments which persist down to 1276 m asl. Considering their similarity with the present day shore line, the m- to km-long linear arrangements of microbial structures following the isobaths and parallel to the shoreline are interpreted as paleoshorelines. Each alignment represents a specific water elevation during periods of stable lower water levels. The development of the cow-pie structures along the shoreline may constitute an initial step in the formation of larger structures, resulting in submersed alignments.



In the submersed area (between 1278.5 and 1274 m asl), microbial deposits are also closely associated with polygonal network where they develop following the edges and occasionally colonize the centre of these polygons. The same polygonal geometry associated with microbial mats interpreted as desiccation structures have been observed in the emersed embayment area. Therefore, the submersed polygons are considered as remnants of desiccation polygons related to past stable low water

levels of the GSL. The regular pattern of polygons and their morphology are indicative of desiccation rather than synaeresis formed under subaqueous conditions (Nichols, 2009). This extensive polygon development requires protracted and high amplitude drop in the lake water level. Their presence and stability shown by old USGS aerial images indicate that they formed prior to 1950 and that they have been stable since then. These polygonal networks were previously documented by Currey (1980) but they were not linked to any type of microbial structures. They are similar to the polygonal structures

identified in Great Basin playas (Neal et al., 1968). Currey (1980) interpreted the GSL polygons as a giant desiccation crack network which may have formed in aerial conditions by the drying out of a sediment that was previously waterlogged (Nichols, 2009). Giant desiccation polygons have been recognized in in the recent shore domain of White Rock Bay (this study; Fig. 5a, d). The present-day polygon position down to 1274 m asl (6 m below the MHWL) implies a low water elevation during their formation. The submersion of polygons following a flooding may trigger the preferential development

of microbial structures at the edges of the polygons, i.e. atop the cracks and then favour the preservation of the polygons. This process can be enhanced by preferential fluid migration through cracks and underlying fractures (Fig. 5c, f; Bouton et al., in press). We hypothesize here that the formation of the desiccation polygons occurs during emersion, but that the establishment of mineralizing microbial deposits associated with the polygons allows the preservation of this peculiar sedimentary geometry through time (Gerdes, 2007 and Bouton et al., in press).

**6 Climate as a major driver of microbial deposit distribution**

The previous results suggest that microbial and sedimentary structures accurately record water level fluctuations. The lake water volume increases in spring and early summer owing to snow melt and rainfall, and decreases in late summer and autumn due to extensive evaporation. Analyses of aerial images spanning the last seven decades, combined with our field observations suggest a rather stable development and position of the observed microbial structures (at least 170 years for a

15 cm thick structure and 22 years for a 2 cm thick crust, assuming a growth rate of 0.88 mm yr$^{-1}$). Annual and short-term (i.e. a few years) fluctuations in the GSL water level cannot account for their development given that the inundation episodes are not long enough, even if the GSL water level fluctuation has reached an amplitude of 7 m since 1963. The origin of the microbial deposits, especially the ancient structures above the MHWL, paleoshorelines and polygonal networks, can be related to older and longer water level fluctuations. The peculiar distribution of the microbial deposits and their duration are

therefore used in the following discussion to suggest a chronological succession of the lake level variations. The lake level has strongly varied since its formation ca. 30 ka BP and its variations are showed in Fig. 7, which combines three different reconstructions taken from the literature (McKenzie and Eberli, 1985; Murchison, 1989 and Patrickson, 2010). The first one



is proposed by Patrickson et al. (2010) and encompasses the whole LB history. This curve presents a low resolution pattern for the LB water level variations based on a compilation of radiocarbon ages. The second reconstruction corresponds to a synthetic curve obtained by Murchison (1989) compiled from numerous works combining different approaches (e.g. sedimentology, geomorphology, archaeology, palynology and stable isotopes) (e.g. Ross, 1973; Rudy, 1973; Currey, 1980;

Currey and James, 1982; Currey et al., 1984; McKenzie and Eberli, 1985). This curve spans from the present to the late LB phase and has been calibrated with radiocarbon dating on various materials (e.g. woods, charcoals, marsh). However, Murchison's curve (1989) reflects punctual events whereas continuous and detailed records are mandatory here. The third reconstruction is from McKenzie and Eberli (1985), who reconstructed the last 5500 years using the $\delta^{18}$O isotope signal from the bulk carbonate content of a sedimentary core obtained near Antelope Island. It shows an apparent correlation between the

oxygen-isotope record of the upper 20 cm of the studied core and the direct measurement of the historic GSL water level fluctuations since 1876. The $\delta^{18}$O fractionation associated with lake elevations yields an average rate of approximately 0.35‰ m$^{-1}$. This approach provides a high temporal resolution with, on average, one extrapolated value of the lake level every 88 years. However, their sampling resolution (ca. 0.5 cm) corresponds to a time-average of ca. 20 years and hence tends to smooth the signal, including several punctual events.

According to McKenzie and Eberli (1985), Murchison (1989) and Patrickson (2010) reconstruction, only the initial history of LB or the last ca. 11.5 ka BP phase correspond to a period of low water level around 1280 m asl (Spencer et al., 1984) and may be involved in the formation of the extended microbial deposits described above. The presence of well-dated microbial deposits on the four main LB terraces indicates that they formed during the whole lake history. However the geomorphology of the successive LB terraces, characterized by a reduced extension of the shore, is conducive to a restricted spatial extension

of the associated microbial deposits.

The role of climate in the GSL level fluctuations was recognized both in its recent history (e.g. Lall and Mann, 1995) and during the LB phase. For instance, Godsey et al. (2005) suggested that very dry conditions resulted in the drastic water level decrease (ca. 160 m) following the Provo stage (ca. 12 ka BP; Fig. 7). McKenzie and Eberli (1985) proposed a climatic control on the GSL level fluctuations. They discussed cyclic fluctuations over the past 5500 years (Fig. 7) and estimated an

amplitude for the water level fluctuations of ca. 6 m, (i.e. between 1278 and 1284 m asl), showing a 1500 year duration for the maxima and a 750 year duration for the minima. Similarly, Murchison (1989) proposed several long-term water level increases and decreases (Fig. 7). Our results suggest that the distribution of the microbial deposits and sedimentary structures (e.g. alignments, polygons) is related to fluctuations in the lake level and consequently, mostly to climatic changes. The local influence of tectonics (Holocene fault activity) cannot be excluded but cannot be related to the post-Bonneville isostatic

rebound, which is considered as negligible during this period (Oviatt, 2015).

Here we propose to discriminate the different major climatic events involved in the non-random distribution of microbial deposits (Fig. 7) using a composite LB curve provided by Patrickson et al. (2010), McKenzie and Eberli (1985) and Murchison (1989). A water elevation of 1283 m asl is necessary to explain the presence of the laminated and cauliflower structures 3 m above the MHWL (Fig. 7) dated ca. 21.9 ka BP. At anytime, during the GSL phase, water level elevation



exceeding 1283 m asl were probably too brief to allow the growth of microbialites. Considering the higher growth rate suggested in this work (15 cm phase growing over 170 years at 0.88 mm yr[-1]), only intervals of more than 150 years might be considered. The unique candidate issued from the low resolution Murchison curve's is an interval between ca. 3.5 to 1.3 ka BP. While the higher resolution curve proposed by McKenzie and Eberli (1985) confirms this high water level episode,

its duration is reduced down to 250 years. Although this time interval is sufficient at a maximal growth rate, the most commonly reported rates (ca. 0.5 mm yr[-1]; e.g. Jahnert and Collins, 2012) do not allow the growth of this microbial structure in such short time interval. Moreover, the 21.9 ka BP radiocarbon age obtained for the laminated and cauliflower microbial structures rules out the hypothesis of a development during the GSL phase and rather support the Stansbury oscillation (22.0 to 20.0 ka BP; Patrickson et al., 2010) as a possible candidate. Observed structures moreover rather resemble to the ones

developed on Stansbury and Provo terraces (21.0 ka BP and 15.5-12.5 ka BP, respectively; Oviatt, 2015) than the ones developed during the GSL phase. However, this novel age is puzzling regarding the published curves of LB water level reconstruction (e.g. Oviatt et al., 1992; Patrickson et al., 2010; Oviatt, 2015). Indeed, the estimation of the lake water level was 50 m higher at this period than the proposed low water level associated with the development of such structures. Their presence in the deeper part of the Bonneville Basin, close to the modern GSL configuration may therefore underline

previously unrecognized low water level occurrence during the Stansbury oscillation (22.0 to 20.0 ka BP; Patrickson et al., 2010).

The development of microbial deposits near the shore or toward the lake may have taken place during the entire history of the GSL and potentially during the late LB phases, because they could have formed near or below the mean range of the water level fluctuations. These deposits indicate a water level for the shoreline close to the modern one during this period. It

fits with episodes above the MHWL recorded in Murchison's curve (1989). However, the presence of desiccation polygons and microbialite-rich shorelines at low elevations (1274 m asl for the deeper desiccation polygons and 1276 m asl for the deepest paleoshoreline) suggests the occurrence of very low lake levels, which are seldom in the history of the GSL (Fig. 7). All of the paleoshorelines displaying from the modern shore to ca. 1276 m asl may have developed between ca. 3.6 – 2.9 ka BP.; Fig. 7). The corresponding low level can explain the shallow polygons but is not low enough to explain the deepest ones

(ca. 1274 m asl). The formation of deeper polygons and paleoshorelines could also have occurred during other major events of low water levels (Fig. 7). The first known event corresponds to the Holocene climatic optimum (ca. 7.5 – 5.0 ka BP.; Currey, 1980), which resulted in a drastic water level drop for North American lakes (Street and Grove, 1979). During this period, the GSL level dropped to 1274 m asl (Murchison, 1989) and a large part of the GSL was an almost desiccated playa (Currey, 1980). Currey (1980) postulated that the desiccation polygons were formed during this period, and Murchison

(1989) further narrowed down the dating for polygon development between ca. 6.9 – 6.0 ka BP (Fig 7). A low lake level during this period has been also suggested using a slight angle truncation in a seismic reflector that correlated with a 6.7 ka BP tephra (Colman et al., 2002). A significant amount of authigenic dolomite in the sediments in association with the playa configuration is also documented for this time interval (McKenzie and Eberli, 1985). Dolomite precipitation can result from chemical precipitation under evaporitic conditions (Pierre et al., 1984) but also from microbial organomineralization





(Vasconcelos et al., 1995). A second and older episode of low water levels occurred at the end of the LB phase, ca. 11.5 ka BP (Fig. 7). A drill core from the deepest part of the GSL (41° 07'50"N - 112° 33'46"W) showed the presence of thenardite layers between 14.4 and 8 m below the lake floor, with an intercalation of black sapropelic muds (Balch et al., 2005). The age of these layers was estimated as between $17 \pm 3$ and $11 \pm 4$ cal ka (U-series geochronology; can be approximated in uncalibrated radiocarbon age to $13.8 - 9.6$ ka BP) and probably relate to a low lake level succeeding the post-Provo regression, and preceding the Gilbert transgressive episode. The deposition of a 6-m-thick evaporitic layer in this part of the lake suggests an extensive desiccation event forming a playa environment with only few remaining salt marshes and/or hypersaline ponds. The lake level during this event was low enough to expose the whole microbialite-polygon area. The presence of a 8-m-thick sedimentary record above the thenardite layer, ca. 25 km away from the NW Antelope Island, in one of the deepest parts of the GSL, suggests that extensive erosion removed the homogeneous sedimentary cover in the NW Antelope Island area. Accordingly, even if the accumulation of sediment was probably thinner near Antelope Island, given that it is a topographic high, the thickness of the sedimentary record is probably still too large and will cover the polygons. Additionally, the homogeneity of the seismic reflectors acquired close to Antelope Island (Colman et al., 2002), suggest a continuous sedimentary record at the lake scale and low erosion. It is confirmed by sedimentological correlations, which indicate at least 4 m of sedimentary record above the tephra layer (estimated ca. 6.7 ka BP) located stratigraphically above the salt deposits (Spencer et al., 1984). Thus, we conclude that the Holocene climatic optimum event remains the most probable phase during which deeper polygons and microbialitic fossil shorelines formed.

## 7 Impact of tectonics on the distribution of the microbial deposits

The spatial distribution of the microbial deposits and sedimentary structures can be an accurate proxy of the water level fluctuations through time and most of these fluctuations can be explained in the climatic record of the LB and GSL. However, the peculiar straight distribution of the Antelope Island microbialites along a sharp topographic fall (Fig. 5a, f) raises the question of whether tectonics may be an additional driver on the development and distribution of microbial deposits. The topographic fall is interpreted as a major fault belonging to the GSL complex fault system with a N-S to NW-SE orientation (Clark et al., 2014). The scattered lens-shaped detachments of microbial structures, which developed along the fault, probably initiated by gliding (Fig. 5g). Glidings are commonly observed in an extensional context (e.g. passive margin) and reflect the response of the superficial sedimentary cover to extensional tectonics (Cobbold and Szatmari, 1991). Colman et al. (2002) reported the presence of possible microbial mound structures on seismic profiles located in the central area of the GSL, some of which are partly buried under Holocene sediments. Their occurrences were confirmed by geophysical prospects (CHIRP, Sidescan sonar; Baskin et al. 2012). They formed atop of (sub) vertical faults (Colman et al., 2002) or on structural microtopographic highs with onlapping sediments on the corresponding hanging wall lows (Baskin et al., 2012).




The tectonic framework in the Great Basin has led to significant groundwater fluxes, especially near the GSL (Cole, 1982). The contribution of hydrothermal and groundwater influxes is limited to 3% of the total water input, but accounts for 18% of the total ionic influx (Hahl and Langford, 1964). The ion-rich groundwater could enhance the mineralization of the microbial structures. A similar relationship between fluid circulation and microbialite formation has already been observed in the East

African Rift where microbialites (chimney stromatolites) were aligned along fractures and resulted from hydrothermal water input (Casanova, 1994). The presence of microbial deposits along the extended fault system in the GSL opens new perspectives with regards to the role of groundwater fluxes in microbialite mineralization processes.

## 8 Conclusions

The Great Salt Lake is a modern hypersaline lake in which an extensive microbial system composed both of microbial mats

and microbialites has developed. Detailed mapping of microbial, non-microbial and sedimentary structures along the western margin of Antelope Island using aerial images and field observations can be used to identify a specific non-random distribution of the microbial deposits and sedimentary structures. Microbial deposits, covering ca. 90 km² of the margin, are aligned with paleoshorelines and synsedimentary normal faults and are frequently observed on the border of large-scale polygonal structures. Climate-driven water level fluctuations are evidenced here as a strong potential candidate to explain the

distribution of the paleoshorelines and polygons associated with the microbialites. Almost all of the lowermost mapped paleoshorelines composed of coalescent microbial deposits may have developed during ca. 3.6 and 2.9 ka BP. The most extended and distal submersed desiccation polygons probably formed during a period of prolonged lowstand (Holocene climatic optimum ca. 6.9 and 6.0 ka BP). Climate is therefore a major driver of the GSL microbial deposits, but the influence of tectonics (normal fault system) is also evidenced. Interestingly, radiocarbon age ca. 21.9 ka BP obtain for the laminated

and cauliflower structures observed up to 1283 m asl question the commonly assumed LB lake level record. These structures suggest a low water lake level episode (close to the modern one) during the Stansbury oscillation, which was not reported in the literature so far. The LB lake level possibly dropped off drastically during the Stansbury oscillation. The spatio-temporal distribution of the microbial deposits therefore appears to be a sensitive marker of environmental changes and tectonics. They can thus provide an additional tool to reconstruct paleoenvironmental and paleoclimatic changes at a regional scale.

The interplay of the contribution of the climate and tectonic frameworks is determinant in the non-random distribution of the microbial deposits along the western Antelope Island margin. A closer observation of these microbial structures reveals a variety of mineral products (Mg clays, aragonite, dolomite) and fabrics. These features cannot be only explained by the external factors discussed above (e.g. climate and tectonics). The combination of observations of environmental conditions (e.g. hydrodynamics, substrate) as well as microbial and physicochemical parameters will eventually lead to an integrated

conceptual model for the development, distribution and diversity of the GSL microbial deposits.



**Acknowledgements**

This work is a contribution to the SEDS team at the Biogéosciences Laboratory (Dijon, France) and Géoressources et Environnement team at the ENSEGID (Bordeaux, France). This study is also supported by funding provided by GDF Suez EP (ENGIE). We acknowledge John Luft and his colleagues from the Utah Division of Wildlife Resources for their help in
the field and the boat provided to sample in the Great Salt Lake. We thank Fabien Garcia and Cédric Bougeault for the XRD analyses. We are grateful to the U.S. Geological Survey, Google Earth, Utah Automated Geographic Reference Center, Garmin and Digitalglobe® for providing the aerial and satellite images used in this paper. We thank NASA and the USGS for providing the Landsat space images. These data are distributed by the Land Processes Distributed Active Archive Center (LP DAAC), located at USGS/EROS, Sioux Falls, SD. http://lpdaac.usgs.gov.

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





**Table 1: Radiocarbon ages acquired for this paper.**

| Lab. number | 14C age BP | 14C error | Material |
| --- | --- | --- | --- |
| POZ-77888 | 21900 | ± 120 | Laminated and cauliflower |
| YAUT-016612 | 10634 | ± 54 | Cow-pie |
| YAUT-016613 | 7312 | ± 113 | Cow-pie |
| YAUT-016611 | 7060 | ± 40 | Cow-pie |
| POZ-77886 | 5780 | ± 40 | Cow-pie |
| YAUT-016617 | 3283 | ± 42 | Ooid |
| YAUT-016620 | 2678 | ± 72 | Domes and columns |
| YAUT-016619 | 64 | ± 96 | Flat pustular crusts |
| YAUT-016618 | -11 | ± 46 | Oncoid |





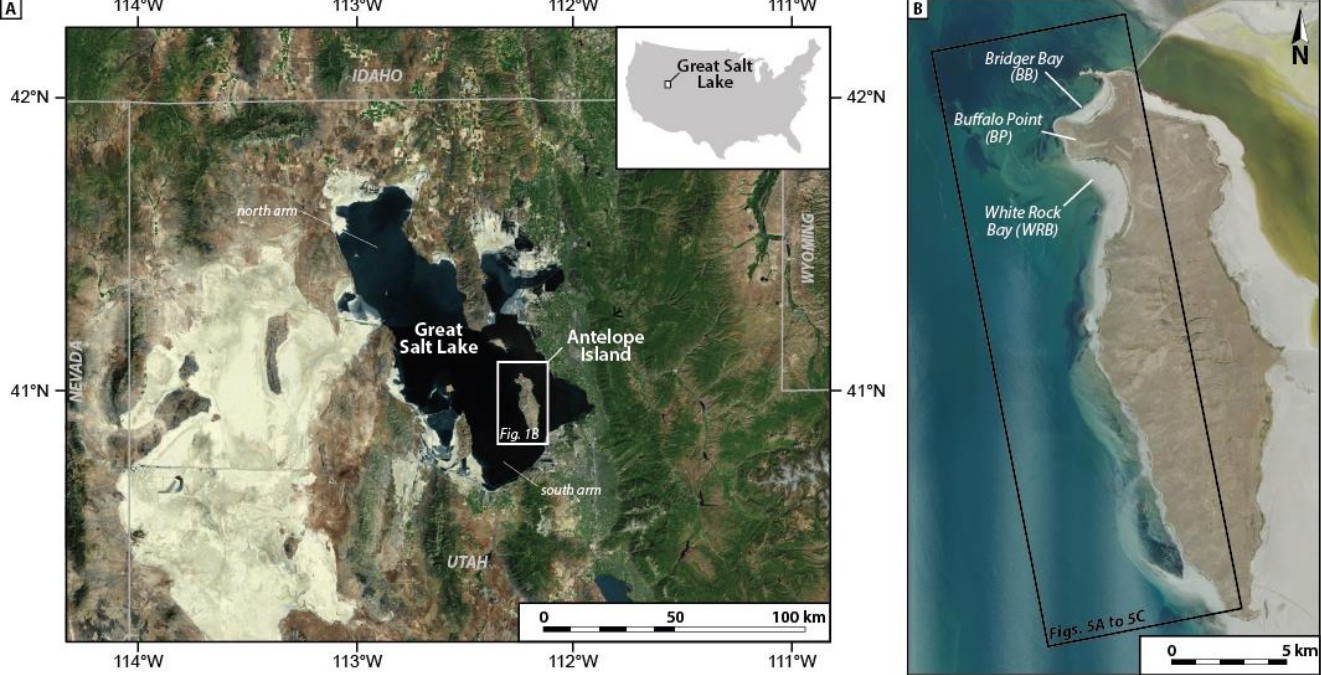

**Figure 1: Satellite images of the Great Salt Lake (A) and Antelope Island (B) showing the studied localities: Bridger Bay (BB), Buffalo Point (BP) and White Rock Bay (WRB). The satellite images are provided by Flashearth (with Bing maps; © Microsoft**
5   **Corporation – Imagery © Harris Corp, Earthstar Geographics LLC), and the Utah Automated Geographic Reference Center (2014 NAIP 1 Meter Orthophotography).**





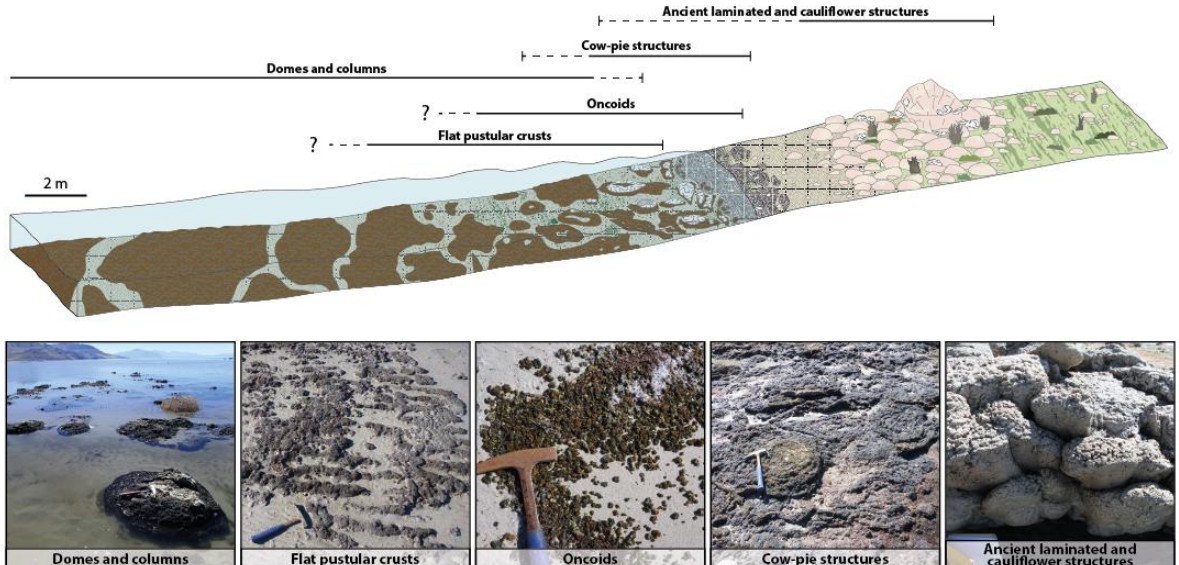

Figure 2: Shore-to-lake transect depicting the position of the microbial macrofabrics used for mapping.



**Figure 3: Panoramic view of the depositional environments on the shore. The microbial deposits are fossilized; the non-microbial deposits are dominated by Cambrian quartzite rounded blocks, quaternary micro-conglomerates and some ooid sands. (B) Details of the microbial deposits located in the area subject to annual water level fluctuations along the shore (2013 shown). The boundary between the living mat and the fossil part of the microbial structure indicates the water level elevation. (C) Panoramic view of the lakeward depositional environment. Most of the microbial deposits (cow-pie structures) consist of a living part (dark brown); the light grey appearance in microbial deposits surface refers to only mineralized structures, suggesting a prolonged period of emersion. (D) Details of the lakeward microbial deposits and the surrounding ooid sand. (E) Relative chronology between the cow-pie structures, dated as at least ca. 10.6 cal ka, onlapping older laminated and cauliflower structures.**





**Figure 4: Polygonal architectures of the emersed White Rock Bay ooid sand flat. (A) Aerial image of the White Rock Bay polygons (Google Earth). (B) Field photograph of the emersed polygonal architecture observed in the sand flat with the rise of the underlying green clays in the cracks. (C) Section through the crack of a polygon showing the rise of the green clays. The different vertical shades of green suggest preferential fluid pathways through the clays. (D) Formation of microbial mat hemispheroids through blistering above the polygons. (E) Cut into a microbial mat hemispheroid showing green pigments which are frequently associated with cyanobacteria. (F) Section within a hemispheroid made of cemented ooid grainstone. The presence of an infra-mm thin red-coloured layer 0.5 mm below the top probably reflects the presence of purple sulfur bacteria. (G) Accumulations of mirabilite (Na2SO4· 10(H2O)) within small depressions (sinkholes) in the vicinity of the polygons. (H) Detail of mirabilite crystals.**



**Figure 5: Detailed maps of the western side of Antelope Island. (A)** Distribution and organization of the microbial deposits and sedimentary structures (2014 NAIP 1 Meter Orthophotography). **(B)** Density map of the distribution of the microbial deposits. **(C)** Bathymetric map showing isopleths of 0.5 m. **(D)** Details of the microbial alignments interpreted as paleoshorelines. **(E)** Details of the desiccation polygons. **(F)** GSL major normal fault and lower limit of the microbial deposits (white dashed line); the microbial deposits are present in the footwall (yellow polygon), but not observed on the hanging wall. **(G)** Details of a lens-shaped detachment of the microbial deposits resulting from gliding (white arrow). **(H)** Photographs taken from the heights of the NW area of Antelope Island of the shore-parallel alignments of the microbial deposits on the shore.



**Figure 6: Illustrations of sediment removal. (A) Typical aerial/satellite images of NW Antelope Island (since 1950) with abundant microbial deposits (2014 NAIP 1 Meter Orthophotography). (B) The same area as shown in panel A showing the burial of microbial deposits in ooid sands (Digitalglobe® images through the Garmin Birdseye software; exact date unknown but probably ca. 2013). (C) and (D) depict a microbial structure that is partially buried in the ooid sand; the outcropping part (C; 3 cm high) is the only visible part; after removal from the surrounding sediment, the structure is 40 cm high (D) and the base of the column shows the attachment to an indurated substrate.**







**Figure 7: Reconstitution curve of the water level at Lake Bonneville and the Great Salt Lake showing potential periods of formation of the fossil microbial structures (brown), paleoshorelines (yellow) and desiccation polygons (red).**