# Peer review of "Linking the distribution of microbial deposits from the Great Salt Lake (Utah, USA) to tectonic and climatic processes"

_Biogeosciences, 2015_

## Referee Comment (RC1) · Anonymous Referee #3 · 2 Jun 2016

Summary: In the associated manuscript, Bouton et al. develop high-resolution distribution maps of microbialites and lacustrine deposits of the Great Salt Lake (GSL), Utah. These maps demonstrate interesting heterogeneous spatial relationships among microbial and sedimentary features, shorelines and faults. Using these data the authors argue that microbialite distributions indicate ancient shorelines and relate these shorelines to lake level curves generated in prior works.

This manuscript presents an impressive relatively high-resolution set of maps that provide crucial insight into the distribution of apparent microbialite structures. I commend the authors in generating beautiful images in this manuscript. The authors' interpretations are broadly backed by the collected data. However, the potential variability

in microbialite genetic mechanisms may complicate the conclusions, particularly the association of microbialites with shorelines.

General Comments: Whereas the desiccation polygons have a straightforward relationship to the shoreline, microbialite relationships are likely more complicated. Indeed the alignment of microbialite structures with the modern shoreline suggests they may be shoreline structures, however, it appears that this orientation coincides with faults as well (the faults probably directly influence the shoreline). Indeed, the authors mention that deeper microbialites occur in association with faults (lines 27-31, pg 12). The faults likely provide a source of Ca2+ to the lake promoting localized carbonate precipitation (as in Mono Lake, CA). This may also explain microbialite distributions along polygon edges as the authors discuss (lines 10-11, pg 8). At any rate, both fault and polygon edge proximal structures can be related to a genetic mechanism for local carbonate precipitation (source of Ca2+). In contrast, a shoreline formation mechanism is not explained. If microbial in nature these structures may be limited to shallow photosynthetic depths, although light can penetrate to significant depths depending on water clarity. In addition, calcium contents are relatively low and photosynthesis does not relieve this burden for carbonate mineral precipitation.

Might is also be possible that microbialites occur at deeper depths that are simply not visible because of water clarity? This would certainly lead to easier identification of shallower (and therefore shoreline) structures.

Specific and Technical Comments:

Pg 1 Lines 19-20: "system" appears twice in same sentence

Line 29: microfabric (singular)

Pg 4 Lines 8-9: AMS undefined. In this context the following "spectrometers" is redundant.

Lines 17-20: "microbially-mediated precipitation" and "trapping and binding" are quite

distinct processes. Trapping and binding does not involve mineralization (precipitation) but rather incorporation of detrital grains. Are both processes leading to microbialite formation?

Lines 20-21: It would be informative to identify which of the microbialite structures (domes, cauliflower, etc) host living microbial mats. Same comment for line 3 pg 8.

Pg 5 Line 1: missing a unit after "few".

Line 2: "…with thicknesses between…"

Line 18: need a "the" before "boulders"

Pg 8 Line 29: The microbialites track approximate shorelines, as they must form in submerged (at least predominantly) environments.

Figure 3: The caption describes a pane E that is not shown.

---

## Author Comment (AC1) · 13 Jun 2016

We thank the anonymous Referee #3 for her/his appreciation of the manuscript and constructive comments. Replies to the raised points are provided below:

General comments: Referee #3 raised the complex relationships between microbialites and geological/geographical structures such as the shorelines. For instance, the presence of cow pie structures is related to a very low water elevation. The observation of present-day microbial structures in the GSL indicates that cow pies are preferentially restricted to the shoreline area and, occasionally, in local topographic highs disconnected from the shoreline (see Bouton et al., 2016, Sedimentary Geology). Therefore, the type of microbial structures and their distribution can be powerful tools to track the

[Figure]

position of the shoreline through time, and thus, of the water lake level.

Some alignments of microbial deposits were interpreted in this manuscript as paleoshorelines. As mentioned by Referee #3, some of these alignments may coincide with faults, depending where observations are made. In NW Antelope Island (Fig. 5C), the alignments strongly follow the crooked geometry of the coast; while they are partly conform with a N135° fault orientation at the western margin (Dinter and Pechmann, 2014). These alignments can be followed northward all along the coast where they differ from the general fault pattern. Both systems (paleoshorelines and fault-dominated alignments) may coincide, but their own characteristics (e.g. topographic drop-off vs. the coastal morphology) allow to discriminate them.

As mentioned, fluid circulation through faults or cracks associated with polygonal networks may provide an important source of $Ca2+$ in the lake, favouring the carbonate precipitation and therefore, the formation of microbialites. However, we also notice that the presence of microbialites in the centre of polygonal structures cannot be explained by fluid transfers from the polygon edges. As a result if faults or cracks may indeed help microbialites formation, the latter is not restricted to this peculiar way of circulating the fluid.

This statement is further supported by the observation that while the $Ca2+$ content is low in the GSL water (ca. 230 mg/l; USGS data), it does not prevent the precipitation of carbonate phase within microbial mats at equilibrium with surface water according to the saturation index of aragonite and dolomite calculated using Phreeqc modelling (see attached figure). GSL waters are supersaturated respectively to these mineral phases allowing precipitation of carbonates everywhere in the lake and not only in relation with fault circulating fluids.

Deep microbial structures have been recognized in the literature (Colman et al., 2002) and acknowledge in this work (Fig. 5C- dashed yellow polygons). However, they are rarer than shallow microbialites. We agree that the clarity of the water may have somewhat altered our mapping of deep structures. Nevertheless, the large studied dataset of aerial and satellite images (continuous since 1972 and throughout different season of the year) repeatedly indicate that deep structures are scarce and patchy.

Specific and technical comments:

We thank the Referee #3 for his/her specific and technical comments. These modifications will be introduced in the next version of the paper.

- Page 1, lines 19-20: the term "system" appeared twice in the same sentence, the first was changed into "lake".

- Page 1, line 29: the term "macrofabric" was put in singular.

- Page 4, lines 8-9: the abbreviation AMS (Accelerator Mass Spectrometry) was defined and the text was changed in consequence.

- Page 4, lines 17-20: we agree that "microbially-mediated precipitation" and "trapping and binding" are different, but they both contribute to the formation of microbialites as they are intrinsically linked. In addition, the historical definition of microbialites by Burne and Moore, (1987) including their formation refers to both processes. A reference to Burne and Moore (1987) was added in the text in that sense.

- Page 4, lines 20-21: we think that it is not relevant to specify here the type of microbialites, which host living microbial mats since these microbial structures are not yet described in the main text. These microbial structures are described just below including information if they host living microbial mats. However, in page 8, line 3, we agree that this information can be helpful to precise the relation between living microbial mats and the different microbialites.

- Page 5, line 1: we changed "just few" by "slightly above" in order to clarify the text.

- Page 5, line 2: we agree with this comment and we modified the text accordingly (plural to "thicknesses").

- Page 5, line 18: we added "the" before boulders.

- Page 8, line 29: We agree with Referee #3's comment and we added the suggested sentence in the text.

- Figure 3: The description of a panel E in the caption is a remnant of a previous version of the manuscript. This will be deleted.

On the behalf of my co-authors, kindest regards.

Anthony Bouton
* * *
[Figure]

[Figure]

**Figure S8**: Results of Phreeqc modelling based on GSL surface water composition (USGS; Table S2). Saturation index as a function of the pH for three minerals (amorphous silica, aragonite and dolomite). Each mineral is represented by two curves showing the variation induced by seasonal temperature fluctuations (M: May; J: July).

**Fig. 1.**

---

## Referee Comment (RC2) · Anonymous Referee #4 · 8 Aug 2016

In this manuscript, the authors analyzed patterns of microbial deposits in Great Salt Lake and set those in relation to potential shaping factors influencing the formation of those patterns, such as climatic and tectonic factors. The introduction provides a good overview of the topic and the state of knowledge, citing relevant literature. The visualization of the described structures with graphs and images is excellent. However, there are several points regarding the presentation and discussion of results that the authors should pay attention to: (i) The research objectives given are very short and general and should be made more specific. What did the authors expect to find? What is their contribution to the state of knowledge in this area of research? (ii) All the descriptions listed in section 3 appear to be own findings of the authors, however, as far as I can

see there is no information about the methods used to obtain these results. Even if this section mainly contains geological and mineralogical descriptions of the different strata and deposits, some basic information about how samples were taken and analyzed would be desirable. If the authors here also build on already existing knowledge published by other authors, this should be pointed out more clearly. In general, the structure is not clear to me. Why is section 4 divided in methods and results and the other sections are not? (iii) In general, the combined results and discussion section appears rather lengthy and should be written more concisely. Each section is focusing on a different aspect, however, the same factors and phenomena are mention again in each section. Here, the most relevant factors should be pointed out more clearly and strongly. Among all the information given, it is often difficult to pick the relevant points that make this study different from previous studies (at other sites). What are the main findings of the authors, and how do their findings contribute to our current understanding of the described processes, going beyond Great Salt Lake? p. 13, l. 22-30: These are strong points and should already be more visible in the preceding sections to highlight more strongly how the integration of these new tools enabled the authors to make a particular finding.

---

## Author Response (AR1)

**Anthony BOUTON**

Université de Bourgogne Franche-Comté
UMR CNRS/UBFC 6282 - Biogéosciences
6 Boulevard Gabriel
21000 Dijon
FRANCE
Tel : +33 (0) 6 82 64 23 56
Email : anthony.bouton@u-bourgogne.fr

Dijon, August 28th 2016

Dear Editor,

Please find herewith the revised version of our manuscript entitled:
"**Linking the distribution of microbial deposits from the Great Salt Lake, (Utah, USA) with tectonic and climatic processes**"

Anthony BOUTON, Emmanuelle VENNIN, Julien BOULLE, Aurélie PACE, Raphaël BOURILLOT, Christophe THOMAZO, Arnaud BRAYARD, Christophe DUPRAZ, Guy DÉSAUBLIAUX and Pieter T. VISSCHER

Corresponding author: Anthony BOUTON

We thank you for your decision. We have carefully considered your comments and those made by reviewers #3 (June) and #4 (August); the manuscript has been modified in consequence. The modifications can be followed through the *MS track and changes* solution.

- *Reviewer #3*

The previously proposed changes in the reply of *Reviewer #3* comments were added in the manuscript. You can find all the details in the discussion reply of the 13th June.

- *Reviewer #4*

The main points raised by *Reviewer #4* were addressed in this revised version of the manuscript.
(i) A sentence was added in the introduction (P2, l22-25) to highlight the significance of this approach and work in this area of research.
(ii) We agree with the comment that the methodology should be better specified. We added a paragraph in the third section (P4, l3-13*)* describing more precisely the database, sampling and the analytical tools. The datasets used in this manuscript are original and were not derived from any other publications.

(iii) *Reviewer #4* pointed out that the structuration of the manuscript does not sufficiently highlight the main findings proposed here. We proposed to slightly restructure the section in order to better emphasize the two main controlling factors, i.e. climate and tectonics. In the previous version of the manuscript, sections 5 and 6 were dealing with the climate influence and section 7 with the tectonics. In this new version of the manuscript, the previous sections 5 and 6 were fused into a new one entitled *5: Role of climatically-induced water level fluctuations on microbial deposits distribution*. The sections and subsections headings were modified in consequence.

The conclusion was also deeply rewritten for the same purpose to highlight the main findings proposed in this manuscript.

Others minor modifications were made to the text, e.g. correcting some spelling issues or adding references of new works recently published.

On behalf of my co-authors, I thank you for your consideration and I am looking forward to your response.

Sincerely yours,
Anthony BOUTON

[revised manuscript text omitted]